# Translation, cultural adaptation, and validation of the PHQ-9 and GAD-7 in Kinyarwanda for primary care in the United States

Frank Müller[1,2,3]*, Alexis Hansen[1], Megan Kube[1,4], Judith E. Arnetz[1], Omayma Alshaarawy[5], Eric D. Achtyes[6‡], Harland T. Holman[1,2‡]

1 Department of Family Medicine, College of Human Medicine, Michigan State University, Grand Rapids, Michigan, United States of America, 2 Corewell Health Family Medicine Residency Clinic, Grand Rapids, Michigan, United States of America, 3 Department of General Practice, University Medical Center Göttingen, Göttingen, Germany, 4 Graduate Medical Education in Psychiatry, Trinity Health Livonia Hospital, Livonia, Michigan, United States of America, 5 Department of Family Medicine, College of Human Medicine, Michigan State University, East Lansing, Michigan, United States of America, 6 Department of Psychiatry, Western Michigan University Homer Stryker M.D. School of Medicine, Kalamazoo, Michigan, United States of America

‡ EDA and HTH share senior authorship.
* muell313@msu.edu

**Data Availability Statement:** All relevant data are within the manuscript and its Supporting information files.

## Abstract

### Background

Depression and anxiety are significant health burdens that greatly impact the quality of life of refugees and migrants. In this study, we have translated and culturally adapted the Patient Health Questionnaire (PHQ-9) and Generalized Anxiety Disorder Screener (GAD-7) into Kinyarwanda and performed a validation study in a United States (US) primary care setting.

### Methods

A committee of seven experts including psychiatric and family medicine providers, health researchers, and trained medical interpreters translated and culturally adapted the PHQ-9 and GAD-7, and incorporated feedback from cognitive interviews with bilingual participants. The translated instruments were then tested in a cross-sectional validation study. Analyses include internal consistency, discriminant validity, principal component analyses, and confirmatory factor analyses.

### Results

Analyses of 119 responses indicated overall good internal consistency with Cronbach's α of 0.85 (PHQ-9) and 0.92 (GAD-7). Both scales showed acceptable factor loadings between 0.44 and 0.90 in the principal component analyses and showed strong correlations with health-related quality of life and depression/anxiety symptoms measured with visual analog scales. Significantly higher scores for PHQ-9 and GAD-7 were shown among participants with known psychiatric conditions.

**Funding:** FM received funding as Peter C. and Pat Cook endowed clinical research fellow for this work. The funding organizations had no role, in designing the study; collecting, analyzing, or interpreting data; or writing or submitting the manuscript.

**Competing interests:** The authors have declared that no competing interests exist.

## Discussion

PHQ-9 and GAD-7 demonstrated commendable applicability for Kinyarwanda-speaking patients in primary healthcare settings in the US. Our instruments can already be used in primary care settings and thus help to mitigate health disparities. Future research should further validate our tool against gold-standard diagnostics in larger, geographically diverse samples.

## 1. Introduction

Depression and anxiety are major global health burdens, affecting approximately 322 million and 264 million people worldwide, respectively [1]. Both conditions can lead to disability [2], mortality [3], reduced quality of life [4], and worsening of conditions like diabetes, hypertension, or heart failure [5–7]. The United States (US) Preventative Services Task Force (USPSTF) has recommended depression screening in primary care for all adults since 2016 [8] and recently expanded this guideline for anxiety screening [9].

Refugee populations, including refugees from Rwanda [10], are particularly affected by mental health problems such as depression, anxiety, post-traumatic stress disorder, and other mental conditions [11–13]. Refugees and migrants often face mental health challenges due to stress exposure to pre-migration experiences such as violence or persecution in their countries of origins, as well as post-migration difficulties including acculturation, language barriers, social isolation, and economic struggles [14]. Also, traumatizing experiences during the migration pathway, such as arduous flights, lack of sanitation, uncertainty during asylum processes, violence and/or human trafficking can have a considerable impact on migrants' health [15].

Western Michigan, including the metropolitan area of Grand Rapids, has become one of the main destinations for refugees and migrants from Rwanda, Uganda, and the Republic of Congo in the US forming a growing community of Kinyarwanda-speaking residents [16]. Research has demonstrated that these populations are often both highly traumatized and less likely to seek mental healthcare [17], further emphasizing the need for high-quality, culturally competent mental health screening.

Validating mental health screening tools for specific linguistic and cultural contexts is crucial to ensure accurate and effective screening, enable informed clinical decisions, and facilitate standardized assessment across different healthcare settings [18]. This is particularly important for vulnerable populations such as refugees and migrants, where culturally and linguistically adapted tools can help reduce misdiagnosis and improve patient engagement in mental health care.

The Patient Health Questionnaire (PHQ-9) is one of the most frequently used mental health screening instruments [19]. While previous research provided a Kinyarwanda translation of the PHQ-9 for use among patients with epilepsy in Rwanda [20], there were concerns regarding potential misunderstandings due to dialectal variations in Kinyarwanda and differences in healthcare contexts between Rwanda and US primary care. Hence, a new comprehensive translation and validation approach, involving cognitive interviews and close collaboration with local Kinyarwanda interpreters, was deemed necessary for our specific study population of Kinyarwanda-speaking refugees and migrants in West Michigan. Furthermore, a tool to screen for clinically relevant anxiety symptoms such as the Generalized Anxiety Disorder Screener (GAD-7) [21] has not been translated and validated in Kinyarwanda.

To address this gap in care, the objectives of this work were to translate and culturally adapt the PHQ-9 and GAD-7 into Kinyarwanda and validate both screeners in a cross-sectional study conducted in primary care clinics in Western Michigan. By publishing these validated instruments as a free resource for clinicians and researchers, we hope to contribute to the efforts to mitigate health disparities among refugee and migrant populations.

## 2. Methods

This study was comprised of two phases: In the first phase, the instruments were translated and culturally adapted using a committee framework approach including cognitive interviews with bilingual participants. In the second phase, a cross-sectional validation study was conducted in primary care clinics in Western Michigan. Results of this cross-sectional part are reported using STROBE guidelines [22]. The completed STROBE checklist as provided as S3 Appendix.

### 2.1. Translation and cultural adaptation

**2.1.1. Approach and translation objectives.** We used the committee-based translation and adaptation framework described by Valdez et al. which includes rigorous steps of (1) planning, (2) parallel translations, (3) review, (4) adjudication, (5) pilot-testing, (6) revision, and (7) documentation, that allowed us to iterate phases as necessary [23]. We further extended this framework by conducting cognitive interviews instead of pilot-testing with bilingual participants and further added an independent back translation as a final quality control before conducting the validation study.

For the translation process, the English versions of the GAD-7 and PHQ-9 were used as sources and subsequently translated and adapted into Kinyarwanda. Kinyarwanda is a Bantu language with different dialectical variants spoken in Rwanda and also in parts of Burundi, the Democratic Republic of the Congo, Uganda, and Tanzania. The aim for the translation and adaptation process agreed upon among all authors was to generate a semantically similar version retaining format, scales, range of response options, and stimulus questions while using plain and familiar language allowing the instruments to be understood by people speaking different dialectal variants and from a diverse educational background.

**2.1.2. Committee members and roles.** The committee-based approach to translation and cultural adaptation uses a continuous feedback and review process within a team of language, culture, and subject matter experts [24]. Our committee consisted of two clinicians (EDA, HTH) trained in family medicine and psychiatry and proficient in the provision of mental health care to refugee patients, a clinician-researcher (FM) additionally trained in psychosomatic medicine providing both clinical and research expertise, one psychiatry resident physician (MK), and one medical student (AH) to provide perspectives from different levels of training, and two trained medical Kinyarwanda interpreters who have been working for the healthcare system as on-site medical interpreters for several years as linguistical and cultural experts. Interpreters were of male and female sex, middle-aged, and both grew up in Rwanda and the US and thus were proficient in both Kinyarwanda and the English language.

The process of translation and cultural adaptation was completed in six teleconference meetings using Zoom video conferencing software (Zoom Video Communications, Inc., San Jose, CA, USA), each lasting over 60 minutes between April and June 2022.

**2.1.3. Planning.** In the planning step, we first conducted a literature review on existing healthcare survey instruments in the Kinyarwanda language to gain insights about potential obstacles that others faced during the translation process. Furthermore, these collected instruments served as a reference in the committee discussions. During the first meeting, the process

of translation and cultural adaptation was discussed. The PHQ-9 and GAD-7 were introduced, and the clinicians described how these instruments were used and results interpreted in everyday practice. Furthermore, the interpreters provided insights into the cultural perception of mental health problems among the Kinyarwanda-speaking refugee community in the US and the relevant impact on the translation and adaptation process. They highlighted that mental health problems are often stigmatized among Kinyarwanda communities, and they expressed their fear that the use of stigmatized expressions would lead to biased responses and should thus be avoided. The interpreters furthermore shared that many in the community feared being labeled as "crazy" or "mad" and were afraid that a suspicious test result would implicate that their "children will be taken away".

**2.1.4. Parallel translations.** After this first meeting, both interpreters independently translated the instruments within two weeks (parallel forward translation). Additionally, clinicians provided paraphrases and elaborations independently on the meaning of each item, e.g. "This item is designed to address the idea of anhedonia, not enjoying things one previously used to enjoy". The results of these first interpretations were compiled in a Microsoft Excel (Microsoft Corp., Redmond, WA) document together with the PHQ-9 version used in the study of Sebera et al. [20] as preparation for the following committee review.

**2.1.5. Review.** After parallel translations were completed, the review process involved the entire team discussing and refining each item to ensure both linguistic accuracy and cultural appropriateness. In the four review meetings, each item was first introduced by the clinicians and explained what role it plays in patients with clinically relevant depression or anxiety symptoms. The interpreters then discussed matching phrasing to concepts and meanings as well as the format and distinction of questions and responses. Especially challenging was the translation of colloquial metaphors in the English version, such as "feeling on edge" and finding expressions that avoid stigma. Here it became apparent, that even basic wording like "mental health" itself would imply "madness" or "craziness" in a direct translation. In the following meetings, the items were further refined and compiled into a first draft, consisting of four pages.

**2.1.6. Adjudication.** Following the review, a separate adjudication was performed by critically examining the items for consistency, clarity, and cultural appropriateness. While most items were signed off for pilot testing some items needed further revisions and modifications of the translation, particularly focusing on nuanced cultural interpretations and potential linguistic ambiguities. Interpreters also checked for contingency, grammar, and spelling and the preliminary version of the entire survey for pilot testing was compiled.

**2.1.7. Pilot.** The pilot testing was conducted as cognitive interviews with n = 5 bilingual English/Kinyarwanda participants (3 male, 2 female), that we recruited through the Michigan State University Department of African Language. All participants were faculty or students of various academic backgrounds located in the US and the Democratic Republic of the Congo. Cognitive interviews were carried out and video- and audio-recorded using Zoom teleconferencing software. The cognitive interviews were meant to assess the cognitive process in answering questions, e.g. if the items were understood (comprehension), and if necessary and relevant information and underlying concepts were retrieved (retrieval). Furthermore, we aimed to gain insights into the process of answer preparation of participants (judgment) and if challenges arose from formatting the answers (response) [25]. The pilot testing thus did not only focus on the survey items themselves but also the handling of the survey as a whole. Participants were first introduced to the purpose of the questionnaire and that it is administered as a standard screening instrument in a family medicine clinic during every patient visit. Then participants were asked to read out loud each item, answer it, then describe their thought process that led them to their final answer [26]. After completing all items, participants were

asked to comment if any items were complex, hard to understand, perceived as odd or disturbing, or contained grammatical or spelling errors. The recorded video of each interview was reviewed by the study team and concerns raised by interviewees were summarized.

**2.1.8. Final revisions and documentation.** A certified external interpreter who was not involved in the research project provided back-translation of both survey instruments. The notes from the cognitive interviews and the results from the back-translation were then discussed in a committee meeting to finalize the interview. The final versions of the translated instruments are included in this manuscript as a (S1 Appendix).

## 2.2. Participant recruitment and setting

Western Michigan, particularly the Grand Rapids area, has become a significant resettlement location for refugees and migrants from Central Africa, including Rwanda and the Democratic Republic of Congo over the past two decades [16]. Approximately 8,000 refugees from Congo are residing in Grand Rapids and 11 Congolese church congregations are located in the area [16]. Many of these refugees speak Kinyarwanda, as it is widely used in both Rwanda and parts of Congo.

Our study was conducted at Corewell Health, a large non-profit managed healthcare organization with 22 hospitals and more than 300 outpatient facilities in Michigan. We employed a convenience sampling method for this study. No randomization was used due to the specific nature of our target population and the clinical setting.

Participants were enrolled in two ways: First, three family medicine clinics in Western Michigan that are established in providing care to the Kinyarwanda community enrolled patients in their clinics during their routine clinic visits. Second, a Kinyarwanda interpreter at the healthcare system was briefed to enroll patients in all other primary care clinics in the health system. Results of the survey were reported to the provider with the notice that the information obtained was based on a not-yet-validated scale and therefore typical thresholds of the screening instruments may not apply. Recruitment took place from 6 June 2022 to 14 August 2023.

Patients were eligible to participate, if (a) their preferred language for the encounter was Kinyarwanda and a language interpreter was needed, (b) patients were aged 18 or older, and (c) patients agreed to participate. Patients who did not meet the inclusion criteria (e.g., those under 18, or those not requiring Kinyarwanda interpretation) were not approached for the study. Patients were excluded from enrollment when they were not able to provide informed consent.

Patients who met the eligibility criteria were approached by clinic staff or interpreters at the participating clinics. The study was explained to them in Kinyarwanda through on-site or video interpreters, and they were invited to participate. Those interested received a study information document in Kinyarwanda that included information about the study's objectives, procedures, the voluntary nature of participation, and the anonymity of collected data. For participants with limited literacy, this information was read aloud by a Kinyarwanda interpreter. Those who agreed to participate provided written consent. Patients did not receive compensation for participating in this study.

## 2.3. Measures

Besides the two translated instruments PHQ-9 and GAD-7, the survey comprised questions on participants' sociodemographic characteristics (gender, age, education, timing of entrance into the US, self-perceived English language proficiency) as well as if the patient has ever been told by a health care provider that they have a mental health problem (such as depression,

anxiety, schizophrenia, bipolar disorder, obsessive/compulsive disorder, a substance use disorder, or other mental health problems). These items were translated in the same process as described above. As illness concepts such as bipolar disorder do not have a direct translation in the Kinyarwanda language, we added a short description highlighting the main symptoms.

Furthermore, the survey contained the Kinyarwanda version of the EQ5D-VAS to measure patients self-rated overall health [27] on a scale from 0 ("worst health you can imagine") to 100 ("best health you can imagine"). Although the Kinyarwanda EQ5D-VAS was thoroughly translated and provided as an official resource through the EuroQol group, we could not find any literature that these instruments have been previously validated in Kinyarwanda. To broadly assess depression and anxiety symptoms in our sample and to compare them to PHQ-9 and GAD-7 items, we added further visual analog scales with the questions "Overall in the last 30 days, how depressed have you felt?" and "Overall in the last 30 days, how anxious have you felt?". Similar scales have been used and validated in other settings [28].

Besides these numerical measures, we also checked each completed survey for annotations made by the respondents.

## 2.4. Sample size

For validation studies, typically an item-to-response ratio from 1:5 to 1:10 is favored [29], suggesting we needed at least 90 responses for the PHQ-9 and 70 responses for the PHQ-7. Our analyzed sample consisted of n = 119 responses, exceeding this minimum threshold.

## 2.5. Statistical analysis

Descriptive statistics including frequencies, percentages, means and standard deviations (SD) were used to characterize our sample and the survey responses. Internal consistency of both scales was assessed with Cronbach's α and McDonald's Ω. Sensitivity analyses for consistency were conducted among various sociodemographic subgroups. For each item of both scales, inter-item and inter-scale correlations were calculated.

After assessing Kaiser-Meyer-Olkin-Criteria (KMO) and performing Bartlett's test of sphericity, principal component analyses (PCA) were used to extract factor loadings. Furthermore, confirmatory factor analyses (CFA) were used to test the known one-factor model of both instruments [30, 31]. Standardized factor loadings were extracted for each item and are displayed as supplemental resources (S2 Appendix).

Items were tested whether they had a suitable explanatory power using item-total correlation thresholds of <0.3 [32] and factor loadings <0.32 [33]. Further, items were checked for inter-item collinearity using a threshold for correlation coefficient > 0.7.

Discriminant validity was assessed to compare the sum scores of participants with and without known psychiatric conditions using the Mann-Whitney-*U* test. Furthermore, Spearman's rho was used to calculate correlation coefficients between both psychometric scales and respective visual analog scales for depression, anxiety, and health-related quality of life (EQ-5D VAS score).

All descriptive and bivariate statistics were carried out using SPSS V29 (IBM Corp., Armonk NY) and the CFA was conducted using AMOS V29 (IBM Corp., Armonk NY).

## 2.6. Research ethics

The study received approval from the Corewell Health Institutional Review Board (Approval #: 2022–075). Only anonymized data were collected. Participants received written information and provided written consent prior to enrollment.

## 3. Results

### 3.1. Sample characteristics

Out of n = 133 recorded responses, 14 were excluded due to missing values resulting in a sample of n = 119. Participants were predominantly female (68.9%) and had a mean age of 40 years (SD 15.5). Almost half of the participants were recruited in the participating three family medicine clinics (49.6%, n = 59), and the remaining n = 60 participants (50.4%) were enrolled in other primary care clinics of the health care system through Kinyarwanda interpreter.

Only 10 (8.9%) participants reported having a known psychiatric condition, of which anxiety disorder (n = 6), PTSD, and Depression (each n = 4) were the most common. Approximately 17.8% of the participants reported having no school education and 42.9% reported speaking English "not at all". Further sociodemographic information is outlined in Table 1.

### 3.2. Psychometrics and item characteristics

Most participants displayed normal depression (PHQ-9 sum score 0–4, 89.9%) and anxiety levels (GAD-7 sum score 0–4, 95%). Sum scores of both parameters exhibited a left-skewed

**Table 1. Participant characteristics.**

| Characteristic* | | Female | Male | Total |
|---|---|---|---|---|
| | | N = 82 | N = 37 | N = 119 |
| | | n (%) | n (%) | n (%) |
| Sociodemographic characteristics | | | | |
| Age | x̄ (SD) | 38.6 (13.6) | 43.3 (19.1) | 40.0 (15.5) |
| | 18–29 years | 21 (26.3) | 9 (26.5) | 30 (26.3) |
| | 30–49 years | 45 (56.3) | 13 (38.2) | 58 (50.9) |
| | 50+ years | 14 (17.5) | 12 (35.3) | 26 (22.8) |
| Education | 0–4 school years | 27 (33.3) | 5 (13.5) | 32 (27.1) |
| | 5–9 school years | 20 (24.7) | 12 (32.4) | 32 (27.1) |
| | 10+ school years | 34 (42) | 20 (54.1) | 54 (45.8) |
| Time in the US | <1 year | 12 (14.6) | 10 (27) | 22 (18.5) |
| | 1–2 years | 10 (12.2) | 8 (21.6) | 18 (15.1) |
| | 2–3 years | 5 (6.1) | 1 (2.7) | 6 (5) |
| | 3–4 years | 8 (9.8) | 4 (10.8) | 12 (10.1) |
| | 4–5 years | 19 (23.2) | 4 (10.8) | 23 (19.3) |
| | >5 years | 28 (34.1) | 10 (27) | 38 (31.9) |
| English proficiency | not at all | 38 (46.3) | 13 (35.1) | 51 (42.9) |
| | Beginner | 27 (32.9) | 11 (29.7) | 38 (31.9) |
| | intermediate | 13 (15.9) | 12 (32.4) | 25 (21) |
| | Advanced | 1 (1.2) | 0 (0) | 1 (0.8) |
| | Fluent | 3 (3.7) | 1 (2.7) | 4 (3.4) |
| Psychiatric condition | | 5 (6.6) | 5 (13.9) | 10 (8.9) |
| Psychometrics# | | | | |
| EQ-5D VAS score | x̄ (SD) | 81.1 (23.8) | 75 (20.6) | 79.3 (22.9) |
| Self-Perceived Anxiety | VAS 0–100 x̄ (SD) | 4.3 (13.9) | 15.7 (27.8) | 10.5 (21.0) |
| Self-Perceived Depression | VAS 0–100 x̄ (SD) | 7.8 (18.6) | 16.8 (24.8) | 7.8 (19.8) |
| PHQ-9 Sum Score | 0–27 x̄ (SD) | 1.7 (2.7) | 2.9 (4.5) | 2.0 (3.4) |
| GAD-7 Sum Score | 0–21, x̄ (SD) | 1.0 (2.6) | 1.4 (3.6) | 1.1 (2.9) |

* Missing age n = 5, Missing education n = 1, psychiatric condition missing n = 7, EQ-5D VAS missing n = 5, Anxiety VAS missing n = 2, Depression VAS missing = 1,
# EQ-5D: Higher scores indicate better self-perceived health, all other scores: Higher scores indicate higher symptom burden.

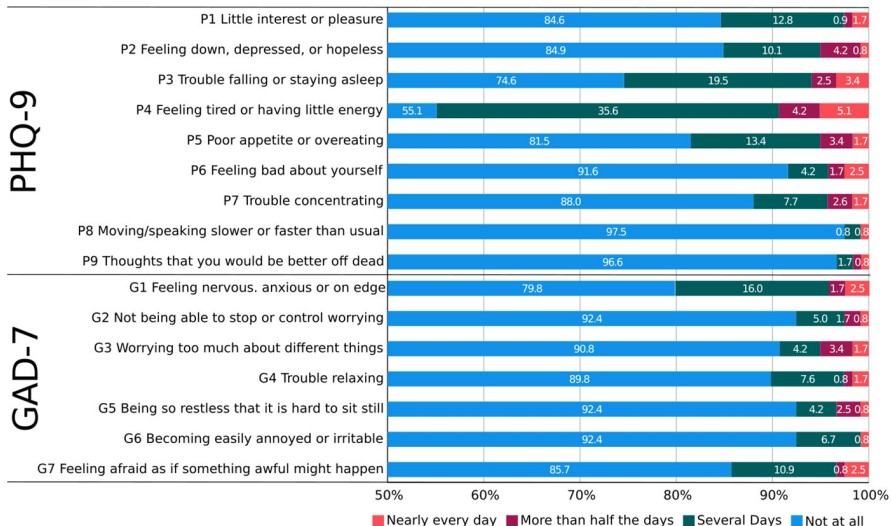

**Fig 1. Participants' responses.**

distribution with a skewness of 3.57 (standard error 0.22) and kurtosis of 18.02 (standard error 0.44) for the PHQ-9 and a skewness of 4.37 (standard error 0.22) and kurtosis of 21.07 (standard error 0.44) for the GAD-7. The distribution of responses is shown in Fig 1.

Inter-item correlation coefficients for the PHQ-9 and GAD-7 ranged between r = 0.06 to r = 0.65, and r = 0.23 to r = 0.65, respectively. A heatmap with correlation coefficients can be found as (S2 Appendix). PHQ-9 and GAD-7 sum scores were positively correlated (r = 0.59, p<0.001).

### 3.3. Reliability

PHQ-9 and GAD-7 displayed overall good internal consistency with a Cronbach's α = 0.85 (McDonald's Ω = 0.87) and α = 0.92 (McDonald's Ω = 0.93), respectively. Item-total correlations ranged between 0.40 and 0.69 for the PHQ-9 and 0.68 and 0.85 for the GAD-7. Cronbach's α did not significantly improve if items were removed from the scale.

In sensitivity analyses, lower α values for both PHQ-9 and GAD-7 were found for older respondents aged 50+ years, respondents who attempted 0–4 years in school, and respondents with lower self-perceived English proficiency and shorter stay in the US (Table 2).

The Kaiser-Meyer-Olkin (KMO) value was 0.760 for PHQ-9 and Bartlett's test of sphericity was significant ($\chi^2$ (36) = 552.53, p<0.001). Similar results were shown for the GAD-7 (KMO value: 0.873, Bartlett's $\chi^2$ (21) = 612.56, p<0.001) suggesting both samples as appropriate for factor analyses. A principal component analysis for a single-factor design revealed factor loadings between 0.44 and 0.80 for the PHQ-9 and 0.76 to 0.90 for the GAD-7 (Table 3).

Using CFA, factor loadings ranged between 0.31 and 0.82 for the PHQ-9 and 0.73 and 0.89 for the GAD-7, however, model fit indices for CFAs were poor (S2 Appendix).

### 3.4. Discriminant validity

Participants with psychiatric comorbidities had significantly higher scores on the PHQ-9 (mean 9.1 [SD 6.7]) compared to those without comorbidity (mean 1.3 [SD 1.8], p<0.001). The same pattern was observed for the GAD-7 scores, with those with comorbidities scoring higher (mean 5.2 [SD 6.3]) than those without (mean 0.6 [SD 1.8], p<0.001) (see Fig 2).

**Table 2. Internal consistency among subgroups.**

| Characteristic | | PHQ-9 | GAD-7 |
|---|---|---|---|
| | | α (95% CI) | α (95% CI) |
| Age group | 18–29 | 0.79 (0.66–0.89) | 0.94 (0.90–0.97) |
| | 30–49 | 0.90 (0.86–0.94) | 0.93 (0.90–0.95) |
| | 50+ | 0.72 (0.52–0.86) | 0.81 (0.68–0.91) |
| Education | 0–4 school years | 0.78 (0.64–0.88) | 0.83 (0.72–0.91) |
| | 5–9 school years | 0.75 (0.60–0.87) | 0.86 (0.77–0.92) |
| | 10+ school years | 0.91 (0.86–0.94) | 0.94 (0.91–0.96) |
| Time in the US | 0–3 years | 0.66 (0.49–0.79) | 0.56 (0.33–0.73) |
| | > 3 years | 0.88 (0.83–0.92) | 0.94 (0.92–0.96) |
| English proficiency | not at all | 0.75 (0.62–0.84) | 0.76 (0.64–0.85) |
| | beginner-fluent | 0.91 (0.88–0.94) | 0.95 (0.93–0.97) |
| Total | | 0.85 (0.81–0.89) | 0.92 (0.89–0.94) |

Self-perceived depression, as measured on a Visual Analog Scale (VAS), exhibited a strong positive correlation with the PHQ-9 sum score (r = 0.53, p<0.001). Similarly, self-perceived anxiety VAS scores showed a strong positive correlation with the GAD-7 sum score (r = 0.65, p<0.001).

Health-related quality of life using the EQ-5D VAS score showed a strong negative correlation with the PHQ-9 sum score (r = -0.59, p<0.001) and a moderate negative correlation with the GAD-7 sum score (r = -0.42, p<0.001).

## 3.5. Comments

Only few participants made written comments or annotations to the survey, mainly elaborating on why participants faced trouble falling asleep (P3), referring to environmental noise due to housing situation or busy roads.

**Table 3. Item characteristics, item-total-correlations, and factor loadings for PHQ-9 (N = 114) and GAD-7 (N = 118).**

| Item | | Score | | Item-Total correlation | α if item deleted | Factor Loading PCA |
|---|---|---|---|---|---|---|
| | | Mean | SD | | | |
| P1 | Little interest or pleasure in doing things. | 0.19 | 0.53 | 0.60 | 0.83 | 0.70 |
| P2 | Feeling down, depressed, or hopeless. | 0.22 | 0.56 | 0.69 | 0.82 | 0.79 |
| P3 | Trouble falling or staying asleep. | 0.34 | 0.68 | 0.59 | 0.84 | 0.66 |
| P4 | Feeling tired or having little energy. | 0.58 | 0.80 | 0.55 | 0.85 | 0.64 |
| P5 | Poor appetite or overeating. | 0.25 | 0.59 | 0.40 | 0.85 | 0.44 |
| P6 | Feeling bad about yourself. | 0.16 | 0.57 | 0.69 | 0.82 | 0.80 |
| P7 | Trouble concentrating. | 0.18 | 0.56 | 0.64 | 0.83 | 0.74 |
| P8 | Moving/speaking slower or faster than usual. | 0.04 | 0.31 | 0.56 | 0.84 | 0.72 |
| P9 | Thoughts that you would be better off dead. | 0.06 | 0.36 | 0.66 | 0.84 | 0.80 |
| G1 | Feeling nervous, anxious or on edge. | 0.27 | 0.62 | 0.68 | 0.92 | 0.76 |
| G2 | Not being able to stop or control worrying. | 0.11 | 0.43 | 0.80 | 0.90 | 0.85 |
| G3 | Worrying too much about different things. | 0.16 | 0.55 | 0.76 | 0.91 | 0.83 |
| G4 | Trouble relaxing. | 0.14 | 0.49 | 0.73 | 0.91 | 0.81 |
| G5 | Being so restless that it is hard to sit still. | 0.12 | 0.46 | 0.85 | 0.90 | 0.90 |
| G6 | Becoming easily annoyed or irritable. | 0.09 | 0.37 | 0.77 | 0.91 | 0.84 |
| G7 | Feeling something awful might happen. | 0.20 | 0.58 | 0.77 | 0.90 | 0.84 |

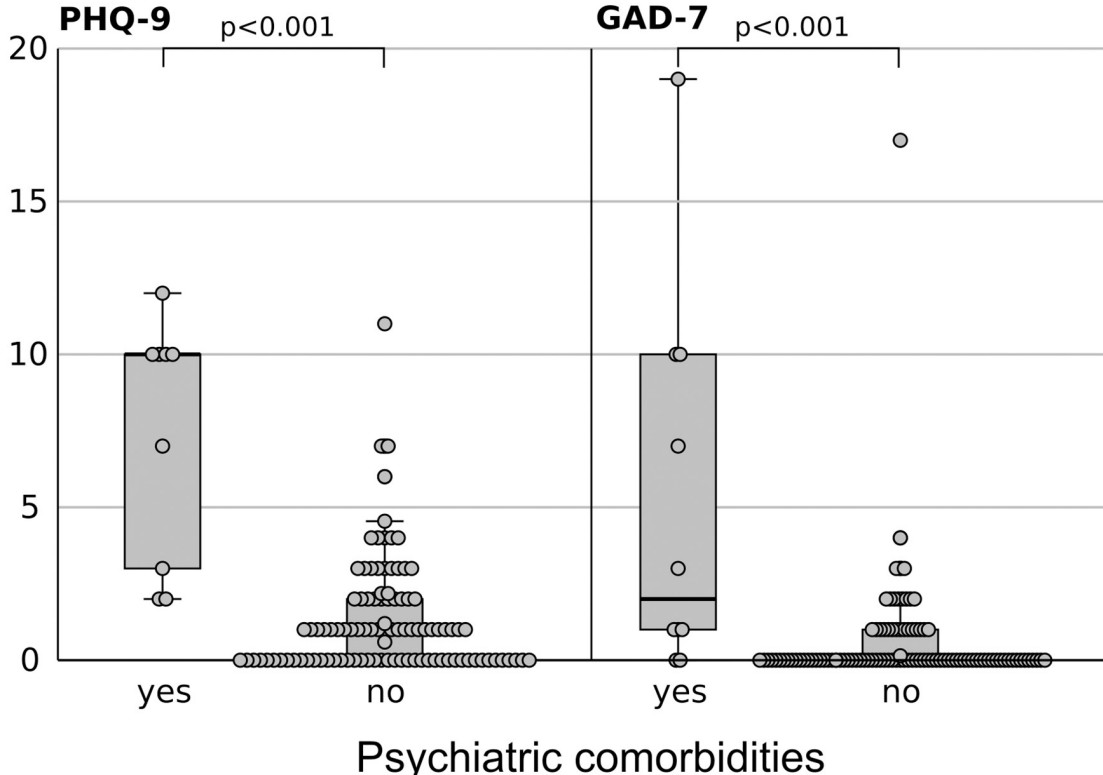

**Fig 2. PHQ-9 and GAD-7 values among participants with preexisting psychiatric conditions (Mann-Whitney-*U* test).**

## 4. Discussion

Studies validating psychometric instruments in a primary care situation for migrants and refugees are limited, especially for smaller language communities. The lack of availability of these instruments, however, is a key factor in insufficient mental health screening among refugee and migrant populations that are particularly affected by the trauma of relocation and mental health problems [34]. Research has demonstrated the practical clinical value of culturally adapted psychometric screening tools in accurately identifying and diagnosing mental health problems among refugee and migrant populations [35].

This work aimed to translate and culturally adapt the commonly used mental health screening instruments PHQ-9 and GAD-7 into the Kinyarwanda language and conduct a validation study by determining internal consistency, factorial structure, and discriminant validity in a cross-sectional sample of migrant and refugee patients seeking care in primary care clinics in the Midwest.

Our study underscores the importance of cultural adaptation beyond mere translation [18, 36]. Our approach to include a diverse committee of translators, clinicians, and researchers familiar with both knowledge in Rwandan culture as well as Western mental health concepts, of which all had hands-on experience in providing health care to Kinyarwanda-speaking communities was crucial to discussing cultural sensitivity. For example, our interpreters highlighted the stigma associated with mental health conditions in Kinyarwanda communities, or the absence of certain disease concepts (such as schizophrenia) leading to careful consideration of our word choice. Similar to other authors, we found that even basic

terms like "mental health" could imply "madness" in direct translation, and pictorial metaphors like "feeling on edge" posed particular challenges [37]. These insights led to adaptations in the phrasing and presentation of items and were crucial in ensuring the PHQ-9 and GAD-7 were not just linguistically accurate, but culturally meaningful and relevant to our target population. Cognitive interviews with bilingual participants provided further insights into how mental health concepts were understood in the Kinyarwanda-speaking community [38].

We found for both instruments an overall good consistency; however, lower consistency levels were observed among patients who were new to the US, had lower English proficiency levels, or were older (50+ years). Particularly the PHQ-9 items on poor appetite/overeating and sleeping problems showed lower factor loadings. However, five items had factor loadings >0.6, suggesting the overall measured factor as reliable [39].

Some participants made annotations to the survey, indicating that their impaired sleep conditions were due to environmental noise exposure that was associated with their housing situation. Research has indicated that newly arrived refugees and migrants are often accommodated in shelters, boarding houses, or substandard apartment buildings that lack privacy and are often exposed to noise that may impair their sleep [40, 41]. Similarly, newly arriving refugees often face considerable challenges in retaining their food-related practices [42, 43], with particular hardships in acquiring goods commonly consumed in their country of origin while facing a limited budget [44]. As a result newcomers to the US often face food insecurity [45], which is often further aggravated by language barriers [46]. These aspects likely contribute to a feeling of decreased appetite, especially among newly arrived migrants without being a clear sign that would indicate an impairment of mental health. This may help explain the lower consistency among newcomers and lower factor loadings of the respective items in this cohort.

The rather small number of incomplete surveys suggests that the survey questions were understood well by various patient groups, including those with limited education. The quality of the translation was confirmed in cognitive interviews.

While we have performed a confirmatory factor analysis, the indicated model of fit indices were poor. It is likely that this is due to a combination of lower factor loadings among some of the previously discussed items and the vulnerability of goodness of fit indices when analyzing smaller samples [47].

Although we were not able to validate our instruments against an established gold standard, we found that respective sum scores correlated well with visual analog scales for health-related quality of life, depression, and anxiety. Furthermore, patients with a known history of mental health problems scored significantly higher for both instruments.

This study comes with further limitations that need to be considered when interpreting the results. Although our sample size met the suggested item-to-response ratios, a considerably larger sample size would have led to more robust findings, especially among subgroups of the sample, and might have yielded better goodness-of-fit indices in the confirmatory factor analyses. The majority of enrolled patients in our sample indicated no or few mental health problems thus reducing variability within our sample. This could be addressed with either a larger sample or a sample that includes more people with such problems, e.g. recruited through mental health providers.

Recently other psychometric instruments in the Kinyarwanda language have been introduced, such as the Beck Depression Inventory [48] or the Mini International Neuropsychiatric Interview [49], however, these were not used in our study as they have not been validated in the respective setting of primary care clinics in the US.

## 5. Conclusion

This study provides the first validated Kinyarwanda versions of the PHQ-9 and GAD-7 for use in US primary care settings, offering essential tools for mental health screening in Kinyarwanda-speaking communities. While these instruments demonstrate good internal consistency and discriminant validity, further research is needed. Future studies should investigate validity against standard diagnostic interviews in larger, more diverse samples, including patients from mental health clinics and psychiatric hospitals. Such research could establish diagnostic accuracy and determine population-specific cut-off scores. Despite these limitations, our culturally adapted PHQ-9 and GAD-7 represent significant steps towards more equitable and culturally competent mental health care for Kinyarwanda speakers in primary care settings.

## Supporting information

**S1 Appendix. PHQ-9 and GAD-7 in Kinyarwanda.**
(DOCX)

**S2 Appendix. Heatmap/CFA.**
(DOCX)

**S3 Appendix. STROBE checklist.**
(DOCX)

**S1 Dataset. PHQ-9.**
(XLSX)

**S2 Dataset. GAD-7.**
(XLSX)

## Author Contributions

**Conceptualization:** Frank Müller, Megan Kube, Omayma Alshaarawy, Eric D. Achtyes, Harland T. Holman.

**Data curation:** Frank Müller.

**Formal analysis:** Frank Müller.

**Investigation:** Alexis Hansen, Megan Kube, Eric D. Achtyes, Harland T. Holman.

**Methodology:** Judith E. Arnetz, Omayma Alshaarawy.

**Project administration:** Frank Müller.

**Supervision:** Judith E. Arnetz.

**Visualization:** Frank Müller.

**Writing – original draft:** Frank Müller.

**Writing – review & editing:** Alexis Hansen, Megan Kube, Judith E. Arnetz, Omayma Alshaarawy, Eric D. Achtyes, Harland T. Holman.

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
