## [Decision Letter · Decision Letter 0]

14 Jun 2024

PONE-D-24-14720Translation, cultural adaption, and validation of the PHQ-9 and GAD-7 in Kinyarwanda for primary care in the United StatesPLOS ONE

Dear Dr. Müller,

Thank you for submitting your manuscript to PLOS ONE. After careful consideration, we feel that it has merit but does not fully meet PLOS ONE’s publication criteria as it currently stands. Therefore, we invite you to submit a revised version of the manuscript that addresses the points raised during the review process. I recommend you perform all the requested changes and follow the Reviewers' recommendations for this manuscript.

We look forward to receiving your revised manuscript.

Kind regards,

Eleni Petkari

Academic Editor

PLOS ONE

Journal Requirements:

Reviewers' comments:

Reviewer's Responses to Questions

**Comments to the Author**

1. Is the manuscript technically sound, and do the data support the conclusions?

Reviewer #1: Yes

Reviewer #2: Yes

2. Has the statistical analysis been performed appropriately and rigorously? 

Reviewer #1: Yes

Reviewer #2: Yes

3. Have the authors made all data underlying the findings in their manuscript fully available?

Reviewer #1: No

Reviewer #2: No

4. Is the manuscript presented in an intelligible fashion and written in standard English?

Reviewer #1: Yes

Reviewer #2: Yes

5. Review Comments to the Author

Reviewer #1: I thank the authors for translating and validating mental health assessment tools for use in a minority group. The lack of appropriate tools has remained a big challenge to assessing and treating of depression in some contexts, thus, the study is relevant. However, the study can benefit from a few adjustments as suggested below.

Abstract. Line 33. Mention the number and category of experts who culturally translated the tools

Abstract Line 44: State the practical implications for the translation and validation

Introduction Line 67: Authors state that the PHQ-9 has been translated and evaluated among patients with epilepsy in Rwanda. My understanding is that the tool was used as a mental health screening tool among patients with epilepsy. If this was the case, then authors should provide a clear justification for a new translation and evaluation among the same population.

Line 70: The significance of validating tools is not well emphasized in the background. A strong case can be built

Methods

Line 80: Authors report using the STROBE guidelines however the checklist is not part of the supporting file

Lines 85-87, authors note ‘All commonly used translation and adaption frameworks entail rigorous steps of planning, parallel translations, review, adjudication, pilot-testing, revision, and documentation, that can be iterated as necessary’. This is a rigorous procedure in translation and cultural adaptions, but in the current study, apart from the translation process, other steps are not comprehensively reported. For example, who participated in the review? Why them? What did they review and what are the outcomes? The strength of adaptation is in the details. Authors need to give all the details of what was done and how it was done across all these processes.

159: The study setting is not explicitly explained. Additional details about the setting would help the reader understand better. More details about the Rwandese community in Michigan can help the reader relate better

Line 172: Authors treat exclusion criteria to be a total opposite of inclusion. Essentially exclusion means that the participant would qualify to participate but because of a certain issues or factor, they get excluded. Being 17 and below means that the participant was not qualified to participate from the onset, thus, that cannot be a criterion for exclusion

Line 178: Did consent happen just by reading the research objectives? Was it written or verbal consent? Secondly, in which language was ‘the paper-pencil survey’ which participants read and agreed to participate? Being a study conducted among a minority group, the consent procedure should be elaborate enough.

Line 203: Authors should report the exact sample size the used

Discussion

Overall, cultural adaptation is not sufficiently discussed. This is also a problem of conceptualization. Authors seem to have simplified translation to represent cultural adaptation. While the former is a process in adaptation, the latter is vast and need to be treated as a comprehensive process. Translation does not automatically represent adaptation although it can be part of the process.

Finally, authors need to include a list of supporting materials at the end of the manuscript

Reviewer #2: Comments to the Authors

I reviewed this paper and found it interesting, particularly considering that depression and anxiety are major public health issues. As such, this manuscript deserves publication. However, the following issues need to be addressed:

Introduction

[1] The first paragraph needs to provide a brief overview of the epidemiology of depression and anxiety, highlighting their importance as public health issues.

[2] The second paragraph needs to justify why mental health issues disproportionately impact refugee populations.

Methods

[3] In the last paragraph of the ‘Translation and Cultural Adaptation’ section, it states that “the notes from the cognitive interviews and the results from the back-translation were then discussed in a final consensus meeting and the survey was finalized." You need to clarify whether both the forward and backwards translators were involved. If not, you need to clarify how you managed any inconsistencies between the forward and backwards translations.

[4] Sampling methods for how the participants were included in the study need to be addressed. Is there any randomisation? If so, what were the random sampling methods employed? If not, then still, the mechanism by which the participants were selected to be included in the study needs to be clarified, including the proportion included in each study setting.

[5] Inclusion and exclusion criteria (page 7): The exclusion criteria normally apply to those who already met the inclusion criteria, which wasn't the case in this manuscript, so once the inclusion criteria applied, none of the exclusion criteria excluded any participants.

[b] Inclusion criterion (a) “…a language interpreter was needed.” Why? In the results section, table 1 indicates 13 intermediate, 1 advanced, and 3 fluent speakers. Does this not conflict with inclusion criterion a? In Table 2, only beginners and not at all are presented. This needs to be sorted out and clarified.

[C] The exclusion criterion (c) seems to state informed consent. But on page 10, you stated that “participants received written information and provided written consent prior to enrolment.”

Discussion

[6] The discussion lacks depth and supporting references.

[7] The "Conclusion" section is missing.

6. PLOS authors have the option to publish the peer review history of their article (what does this mean?). If published, this will include your full peer review and any attached files.

Reviewer #1: No

Reviewer #2: **Yes: **Getahun Kebede Beyera

---

## [Author Response · Author response to Decision Letter 0]

6 Aug 2024

Reviewer #1

 I thank the authors for translating and validating mental health assessment tools for use in a minority group. The lack of appropriate tools has remained a big challenge to assessing and treating of depression in some contexts, thus, the study is relevant. However, the study can benefit from a few adjustments as suggested below.

Thank you for your time and efforts in reviewing our manuscript. 

Abstract. Line 33. Mention the number and category of experts who culturally translated the tools

Thank you for this remark. We have added the respective information: 

“A committee of seven experts including psychiatric and family medicine providers, health researchers, and trained medical interpreters translated and culturally adapted the PHQ-9 and GAD-7, and incorporated feedback from cognitive interviews with bilingual participants.”

Abstract Line 44: State the practical implications for the translation and validation

Thank you for your comment. We have refined this section which now reads:

“Our instruments can already be used in primary care settings and thus help to mitigate health disparities. Future research should further validate our tool against gold-standard diagnostics in larger, geographically diverse samples.”

Introduction Line 67: Authors state that the PHQ-9 has been translated and evaluated among patients with epilepsy in Rwanda. My understanding is that the tool was used as a mental health screening tool among patients with epilepsy. If this was the case, then authors should provide a clear justification for a new translation and evaluation among the same population.

There are several reasons why we have decided to translate and validate the PHQ-9:

The authors of the other PHQ-9 validation study used an already translated PHQ-9 and only briefly commented on how their PHQ-9 instruments have been translated. It appeared to us, that the methodology used for the translation can be prone to misunderstandings, especially for people speaking one of the various dialectal variants of Kinyarwanda. We reached out to the authors of the respective articles and had a chance to review the used instrument together with our multilingual team which confirmed our assumption. For example, the translated scale to indicate the frequency of symptoms was identified as a source for misunderstandings (for example, the response option “more than half of the days” can be misinterpreted as ”more than half of the day” and thus lead to biased responses). For this reason, we have decided to apply a more comprehensive translation approach, e.g. by additionally conducting cognitive interviews. 

Furthermore, it can be assumed that the care-setting of people with Epilepsy in Rwanda structurally differs from Kinyarwanda-speaking refugees and migrants seeking primary care in the US, e.g. comorbidities, exposure to stressors, access to health care and many different aspects. Thus, validation results are not automatically transferable between both settings. To account for our specific setting, we worked together with our trained local Kinyarwanda interpreters as suggested in the literature (Schaffer, Bryan S., and Christine M. Riordan. "A review of cross-cultural methodologies for organizational research: A best-practices approach." Organizational research methods 6.2 (2003): 169-215.)

We have added these aspects into the introduction section that now reads:

“The Patient Health Questionnaire (PHQ-9) is one of the most frequently used mental health screening instruments [15]. While previous research provided a Kinyarwanda translation of the PHQ-9 for use among patients with epilepsy in Rwanda [16], there were concerns regarding potential misunderstandings due to dialectal variations in Kinyarwanda and differences in healthcare contexts between Rwanda and US primary care. Hence, a new comprehensive translation and validation approach, involving cognitive interviews and close collaboration with local Kinyarwanda interpreters, was deemed necessary for our specific study population of Kinyarwanda-speaking refugees and migrants in West Michigan.”

Line 70: The significance of validating tools is not well emphasized in the background. A strong case can be built

Thank you for raising this important aspect. We have added a brief emphasis on the significance of tool validation in the background section:

“Validating mental health screening tools for specific linguistic and cultural contexts is crucial to ensure accurate and effective screening, enable informed clinical decisions, and facilitate standardized assessment across different healthcare settings [15]. This is particularly important for vulnerable populations such as refugees and migrants, where culturally and linguistically adapted tools can help reduce misdiagnosis and improve patient engagement in mental health care.”

Methods

Line 80: Authors report using the STROBE guidelines however the checklist is not part of the supporting file

Thank you for this remark. We have added a completed checklist as supplemental file (S3 Appendix: Strobe Checklist).

Lines 85-87, authors note ‘All commonly used translation and adaption frameworks entail rigorous steps of planning, parallel translations, review, adjudication, pilot-testing, revision, and documentation, that can be iterated as necessary’. This is a rigorous procedure in translation and cultural adaptions, but in the current study, apart from the translation process, other steps are not comprehensively reported. For example, who participated in the review? Why them? What did they review and what are the outcomes? The strength of adaptation is in the details. Authors need to give all the details of what was done and how it was done across all these processes.

Thank you for raising this important concern. Our translation process adhered to the steps and the taxonomy of translation frameworks outlined by Valdez et al., including planning, parallel translations, review, adjudication, pilot-testing, revisions, and documentation. The "review" is not to be mixed up with an initial literature review (which is part of the planning phase). It is however a specific part in this translation framework and has involved our team of experts (including clinicians, researchers, and trained medical interpreters) reviewing and refining the initial parallel translations into a single document. This step was crucial for addressing challenges such as translating colloquial metaphors and avoiding stigmatizing language.

We have clarified in the manuscript, that we followed the approach by Valdez et al. and have introduced corresponding subheadings indicating precisely the taken actions and the results of each stage. Furthermore, we now provide a rationale for the setup of our committee and provide more detailed information on the adjudication phase.

Line 159: The study setting is not explicitly explained. Additional details about the setting would help the reader understand better. More details about the Rwandese community in Michigan can help the reader relate better

Thank you for this comment. We agreed and added a paragraph on the Kinyarwanda speaking community in West Michigan:

“Western Michigan, particularly the Grand Rapids area, has become a significant resettlement location for refugees and migrants from Central Africa, including Rwanda and the Democratic Republic of Congo over the past two decades [13]. Approximately 8,000 refugees from Congo are residing in Grand Rapids and 11 Congolese church congregations are located in the area [13]. Many of these refugees speak Kinyarwanda, as it's widely used in both Rwanda and parts of Congo. 

Our study was conducted at Corewell Health, a large non-profit managed healthcare organization with 22 hospitals and more than 300 outpatient facilities in Michigan. Participants were enrolled in two ways: First, three family medicine clinics in Western Michigan that are established in providing care to the Kinyarwanda community enrolled patients in their clinics during their routine clinic visits. Second…”

Line 172: Authors treat exclusion criteria to be a total opposite of inclusion. Essentially exclusion means that the participant would qualify to participate but because of a certain issues or factor, they get excluded. Being 17 and below means that the participant was not qualified to participate from the onset, thus, that cannot be a criterion for exclusion

Thank you for this important remark. We agree and changed the paragraph as follows:

“Patients were eligible to participate, if (a) their preferred language for the encounter was Kinyarwanda and a language interpreter was needed, (b) patients were aged 18 or older, and (c) patients agreed to participate. Patients who did not meet the inclusion criteria (e.g., those under 18, or those not requiring Kinyarwanda interpretation) were not approached for the study. Patients were excluded from enrollment when they were not able to provide informed consent.”

Line 178: Did consent happen just by reading the research objectives? Was it written or verbal consent? Secondly, in which language was ‘the paper-pencil survey’ which participants read and agreed to participate? Being a study conducted among a minority group, the consent procedure should be elaborate enough.

Thank you for this comment. You are right. We have further outlined the enrollment procedure:

“Patients who met the eligibility criteria were approached by clinic staff or interpreters at the participating clinics. The study was explained to them in Kinyarwanda through on-site or video interpreters, and they were invited to participate. Those interested received a study information document in Kinyarwanda that included information about the study’s objectives, procedures, the voluntary nature of participation, and the anonymity of collected data. For participants with limited literacy, this information was read aloud by a Kinyarwanda interpreter. Those who agreed to participate provided written consent. Patients did not receive compensation for participating in this study.”

Line 203: Authors should report the exact sample size the used

Thank you for this remark. We have added this information both in the “Sample Size” paragraph and in the result section:

“Our analyzed sample consisted of n=119 responses, exceeding this minimum threshold.”

“Out of n=133 recorded responses, 14 were excluded due to missing values resulting in a sample of n=119.”

Discussion

Overall, cultural adaptation is not sufficiently discussed. This is also a problem of conceptualization. Authors seem to have simplified translation to represent cultural adaptation. While the former is a process in adaptation, the latter is vast and need to be treated as a comprehensive process. Translation does not automatically represent adaptation although it can be part of the process.

Thank you for highlighting this very important aspect. We have added the following paragraph to our discussion:

“Our study underscores the importance of cultural adaptation beyond mere translation [32]. Our approach to include a diverse committee of translators, clinicians, and researchers familiar with both knowledge in Rwandan culture as well as Western mental health concepts, of which all had hands-on experience in providing health care to Kinyarwanda-speaking communities was crucial to discussing cultural sensitivity. For example, our interpreters highlighted the stigma associated with mental health conditions in Kinyarwanda communities, or the absence of certain disease concepts (such as schizophrenia) leading to careful consideration of our word choice. Similar to other authors, we found that even basic terms like "mental health" could imply "madness" in direct translation, and pictorial metaphors like "feeling on edge" posed particular challenges [33]. These insights led to adaptations in the phrasing and presentation of items and were crucial in ensuring the PHQ-9 and GAD-7 were not just linguistically accurate, but culturally meaningful and relevant to our target population. Cognitive interviews with bilingual participants provided further insights into how mental health concepts were understood in the Kinyarwanda-speaking community [34].”

Finally, authors need to include a list of supporting materials at the end of the manuscript

Thank you for this remark. We have amended various supplemental materials to our submission. These are (1) a Heatmap of correlation coefficients of psychometric items, (2) CFAs Model Charts for both PHQ-9 and GAD-7, (3) CFA Model Fit indices, [Appendix S2] (4) the entire PHQ-9 and GAD-7 instrument in Kinyarwanda language used in our study with corresponding English translations in light grey font [Appendix S1], Strobe Checklist [S3] and (5) minimal datasets of both PHQ-9 [S4] and GAD-7 [S5] items of our cross-sectional study. We believe that this information is sufficient to allow a replication of our study. 

A corresponding list has been added to the manuscript.

Reviewer #2:

I reviewed this paper and found it interesting, particularly considering that depression and anxiety are major public health issues. As such, this manuscript deserves publication. However, the following issues need to be addressed:

Thank you very much for your valuable comments that helped us to considerably improve our manuscript!

Introduction

[1] The first paragraph needs to provide a brief overview of the epidemiology of depression and anxiety, highlighting their importance as public health issues.

Thank you for this remark. We have added this important information:

“Depression and anxiety are major global health burdens, affecting approximately 322 million and 264 million people worldwide, respectively [1]. Both conditions can lead to disability [2], mortality [3], reduced quality of life [4], and worsening of conditions like diabetes, hypertension, or heart failure [5–7].”

[2] The second paragraph needs to justify why mental health issues disproportionately impact refugee populations.

Thank you for this comment. We have added a paragraph highlighting psychosocial stress of refugee populations:

“Refugees and migrants often face mental health challenges due to stress exposure to pre-migration experiences such as violence or persecution in their countries of origins, as well as post-migration difficulties including acculturation, language barriers, social isolation, and economic struggles [14]. Also, traumatizing experiences during the migration pathway, such as arduous flights, lack of sanitation, uncertainty during asylum processes, violence and/or human trafficking can have a considerable impact on migrants’ health [15].”

Methods

[3] In the last paragraph of the ‘Translation and Cultural Adaptation’ section, it states that “the notes from the cognitive interviews and the results from the back-translation were then discussed in a final consensus meeting and the survey was finalized." You need to clarify whether both the forward and backwards translators were involved. If not, you need to clarify how you managed any inconsistencies between the forward and backwards translations.

Thank you for raising this aspect. Indeed, the final consensus meeting was a “committee meeting” including all committee members and thus also our cultural and language experts that provided independent forward translations. We have added this in the respective paragraph that now reads:

“The notes from the cognitive interviews and the results from the back-translation were then discussed in a committee meeting to finalize the interview. The final versions of the translated instruments are included in this manuscript as a supplemental file.”

[4] Sampling methods for how the participants were included in the study need to be addressed. Is there any randomisation? If so, what were the random sampling methods employed? If not, then still, the mechanism by which the participants were selected to be included in the study needs to be clarified, including the proportion included in each study setting.

Thank you for this remark. We employed a convenience sampling method for this study. No randomization was used due to the specific nature of our target population and the clinical setting. We have furthermore added information of the prop

---

## [Decision Letter · Decision Letter 1]

1 Sep 2024

Translation, cultural adaptation, and validation of the PHQ-9 and GAD-7 in Kinyarwanda for primary care in the United States

PONE-D-24-14720R1

Dear Dr. Müller,

We’re pleased to inform you that your manuscript has been judged scientifically suitable for publication and will be formally accepted for publication once it meets all outstanding technical requirements.

Kind regards,

Eleni Petkari

Academic Editor

PLOS ONE

Additional Editor Comments (optional):

Reviewers' comments:

Reviewer's Responses to Questions

**Comments to the Author**

1. If the authors have adequately addressed your comments raised in a previous round of review and you feel that this manuscript is now acceptable for publication, you may indicate that here to bypass the “Comments to the Author” section, enter your conflict of interest statement in the “Confidential to Editor” section, and submit your "Accept" recommendation.

Reviewer #1: All comments have been addressed

Reviewer #2: All comments have been addressed

2. Is the manuscript technically sound, and do the data support the conclusions?

Reviewer #1: Yes

Reviewer #2: Yes

3. Has the statistical analysis been performed appropriately and rigorously? 

Reviewer #1: Yes

Reviewer #2: Yes

4. Have the authors made all data underlying the findings in their manuscript fully available?

Reviewer #1: Yes

Reviewer #2: Yes

5. Is the manuscript presented in an intelligible fashion and written in standard English?

Reviewer #1: Yes

Reviewer #2: Yes

6. Review Comments to the Author

Reviewer #1: (No Response)

Reviewer #2: The authors have addressed all the provided comments and have made satisfactory revisions, and the manuscript is now much improved.

7. PLOS authors have the option to publish the peer review history of their article (what does this mean?). If published, this will include your full peer review and any attached files.

Reviewer #1: No

Reviewer #2: **Yes: **Dr. Getahun Kebede Beyera

---

## [Editor Report · Acceptance letter]

6 Sep 2024

PONE-D-24-14720R1 

PLOS ONE

Dear Dr. Müller, 

I'm pleased to inform you that your manuscript has been deemed suitable for publication in PLOS ONE. Congratulations! Your manuscript is now being handed over to our production team.

Kind regards, 

on behalf of

Dr. Eleni Petkari 

Academic Editor

PLOS ONE